# Investigations on segmentation-based fractal texture for texture classification in the presence of Gaussian noise

Shamik Tiwari[1], Akhilesh Kumar Sharma[2]*, Izzatdin Abdul Aziz[3]*, Deepak Gupta[4], Antima Jain[5], Hairulnizam Mahdin[6], Senthil Athithan[7], Rahmat Hidayat[8]

**1** School of Computer Science & Engineering, IILM University, Gurugram, India, **2** Department of Data Science & Engineering, School of Information Security & Data Science, Manipal University Jaipur, Jaipur, Rajasthan, India, **3** Center for Research in Data Science (CeRDaS), Computer and Information Science Department (CISD), Universiti Teknologi PETRONAS (UTP), Seri Iskandar, Perak Darul Ridzuan, Malaysia, **4** Department of Computer Science and Engineering, Institute of Technology & Management, Gwalior, India, **5** School of Computer Science & Engineering, VIT University, Bhopal, India, **6** Faculty of Computer Science and Information Technology, Universiti Tun Hussein Onn Malaysia, Batu Pahat, Johor, Malaysia, **7** Koneru Lakshmaiah Education Foundation, Vaddeswaram, Guntur, Andhra Pradesh, India, **8** Department of Information Technology, Politeknik Negeri Padang, Padang, Sumatera Barat, Indonesia

* akhileshkumar.sharma@jaipur.manipal.edu (AKS); izzatdin@utp.edu.my (IAA)

**Data Availability Statement:** The data used in our study are owned by a third party and are available from this source: https://kylberg.org/kylberg-

## Abstract

Texture is a significant component used for several applications in content-based image retrieval. Any texture classification method aims to map an anonymously textured input image to one of the existing texture classes. Extensive ranges of methods for labeling image texture were proposed earlier. However, computing the performance of these methods in the presence of various degradations is always an open area of discussion. Image noise is always a dominant factor among various image degradation factors, affecting the performance of these methods and making texture classification challenging. Therefore, it is essential to investigate the interpretation of these methods in the presence of prominent degradation factors such as noise. Applications for Segmentation-Based Fractal Texture Features (SFTF) include image classification, texture generation, and medical image analysis. They are beneficial for examining textures with intricate, erratic patterns that are difficult to characterize using conventional statistical techniques accurately. This paper assesses two texture feature extraction methods based on SFTF and statistical moment-based texture features in the presence and absence of Gaussian noise. The SFTF and statistical moments-based handcrafted features are passed to a multilayer feed-forward neural network for classification. These models are evaluated on natural textures from Kylberg Texture Dataset 1.0. The results show the superiority of segmentation-based fractal analysis over other approaches. The average accuracy rates using the SFTF are 99% and 97% in the absence and presence of Gaussian noise, respectively.

texture-dataset-v-1-0/. The authors did not receive any privileged access to the data.

**Funding:** This research work is supported and funded by the Yayasan UTP grant: 015MD0-166 with title "Data Analytics and Visualization Development System for Subsurface Co2 Storage and Fluid Production", under the Center for Research in Data Science(CerDaS), Universiti Teknologi PETRONAS, Malaysia.

**Competing interests:** The authors state that they do not have any contending financial interests or personal relationships that could have acted to impact the effort described in this work.

## 1. Introduction

Texture classification has been one of the most attractive research areas in the information retrieval domain, which deals with the discrimination of textures. In a computer vision system, texture is the critical characteristic through which an image can be recognized. Due to this characteristic, texture classification plays a vital role in pattern recognition and computer vision. Images can be from medicine, industry, satellite, etc. Since various classifiers exist, the main challenge is designing an effective classifier model with suitable feature extraction from a given textured image [1]. Many classifier models have been proposed with features such as statistics [2], Gabor filters [3], etc. Images are acquired to record or visualize valuable information. However, the captured image may embody a blurred version of the original scene due to inadequacies in the imaging and recording processes. An extensive range of diverse degradations is required to be considered, such as noise impositions, geometrical factors, illumination, color imperfections, and blurring of images [4]. Among all these, noise is one of the most common degradation factors in images, degrading the classifier's performance heavily. Generally, image noise is a significant factor that can influence the performance of a texture classification system.

Content-based image retrieval is a dominant research area when millions of images are available for search in the database. Texture-based information retrieval has been used successfully by researchers with other features such as shape and color [5, 6]. There are varieties of algorithms for texture feature detection. A few prominent methods are discussed here. The extraction of features like those that contrast, uniformity, and entropy from the Gray Level Co-Occurrence Matrix (GLCM) for feature detection has been suggested by various authors during the last decades. Rotation invariant features have offered Local Binary Pattern (LBP) for texture classification [1]. LBP has proven to be the most computationally efficient, high-performance texture feature, but these features are susceptible to noise and are incapable of extracting macrostructure textures. Many authors have utilized local image features, for example, boundaries, edges, and blobs, for texture categorization [7]. Authors offer Scale Invariant Feature Transform (SIFT) for different applications [8]. SIFT descriptors are invariant to scaling, translation, and rotation. Speedup Robust Features (SURF) features are also used to identify images' local features [9]. SURF features are calculated in terms of key points, established by describing the intensity distribution of pixels. Region-based methods are used to partition the image pixels into groups conforming to logical image qualities like illumination, color, edges, sharpness, and texture [10]. Local Binary Patterns have developed as the most noticeable and extensively considered local texture descriptors. Pan et al. [9] have proposed a low-dimensional FbLBP that can be quickly constructed without needing parameter tuning for different databases. In this work, 4 texture databases, namely CUReT, Outex, XU_HR, and UIUC are used, and the proposed FbLBP-based method has achieved more than 10% improvement relative to conventional LBP and 1–3% improvement close to the finest classification accuracy among other LBP variants. Dong et al. [11, 12] have suggested a local descriptor for texture classification termed extremal pattern and locally directional (LDEP). LDEP fetches the extremum location pattern (ELP), extremum compression pattern (ECP), extremum difference pattern (EDP), and directional local difference count pattern (DLDCP), from the sampling points and the neighbor's extremum related local pattern (NERLP). The experiment is conducted on four texture databases, namely Brodatz, Prague, CUReT, Kth-tips2-a, Stex, and UIUC. The results have demonstrated that the proposed LDEP descriptor can attain comparable accuracy in accurate classification rates in diverse conditions such as rotation, viewpoint variation, noise, scale variation, and illumination with other prominent texture classification techniques.

In the last decade, deep neural networks have gained popularity due to their ability to learn in supervised and unsupervised modes. Various authors offered notable contributions for

texture classification using different deep learning models such as Convolutional Neural Network (ConvNet) [13], Capsule Network (CapsNet) [14], fused ConvNet (TexFusionNet) [15], Contourlet Convolutional Neural Network (C-CNN) [16], Bilinear Convolutional Neural Network (BCNN) [17] and Texture CNN (T-CNN) [18]. Researchers have provided many solutions in the past decade for texture classification using deep learning models. However, training a deep convolutional neural network from scratch is challenging since it requires a large amount of labeled training data and considerable experience to ensure appropriate convergence. On the other hand, the methods based on feature engineering must be evaluated in the presence of noise and other degradation factors.

In this work, the impact of noise is explored once the descriptor parameters have been improved for an image dataset. Additive white Gaussian noise is employed as the noise model. A sample increases each pixel's intensity from a Gaussian distribution. This noise model is highly suited to describe thermal noise in CCD and CMOS sensors, which are image sensors. This paper explores the performance of two different texture classification models in the absence and presence of Gaussian noise. These classification models are based on statistical moment-based features and segmentation-based fractal texture features (SFTF). These features are used to design the classification models with a feed-forward neural network. The contribution of the work is to:

- Offer a texture classification method based on segmentation-based fractal texture features.

- Compare the proposed methods with statistical moment-based features.

- Evaluate the method's robustness in significant Gaussian noise in texture images.

The remaining paper is structured into four sections. Material and methods are presented in Section 2. It is followed by Sections 3 and 4, which contain results and discussion, respectively. Conclusive remarks and future scope are given in Section 5.

## 2. Materials and methods

### 2.1 Models

Mathematical models are essential for real-world image formation and degradation processes. The following subsections discuss the image degradation model and noise model.

**2.1.1 Image degradation model.** An input intensity distribution is transformed into an output intensity distribution during the image formation. The input distribution signifies the true (ideal) image, which is not directly accessible but which is required to recover or at least approximate by suitable action over the degradation. In a two-dimensional linear imaging system, the association between the input intensity distribution and the measured output intensity distribution g(x, y) is denoted as a linear superposition integral provided by Eq 1 [19].

$$g(x,y) = \int_{-\infty}^{\infty} \int_{-\infty}^{\infty} f(x',y')h(x-x',y-y')dx'dy' + \eta(x,y) \tag{1}$$

Here, the term $h(x-x',y-y')$ is the linear point spread function, also called impulse response, and $\eta(x,y)$ is additive noise. The above equation can be expressed as convolution for Linear Shift Invariant (LSI) systems, as mentioned below in Eq 2.

$$g(x,y) = f(x,y)*h(x,y) + \eta(x,y) \tag{2}$$

Here, $g(x,y)$ is the blurred image, $f(x,y)$ is the uncorrupted true image, $h(x,y)$ is the point spread function (PSF) that produced the blurring, and $\eta(x,y)$ is the additive noise. All these

terms are in the spatial domain. The convolution operator in the spatial domain is replaced by the multiplication operator in the frequency domain, so Eq 2 can be written as given in Eq 3.

$$G(u, v) = F(u, v)H(u, v) + N(u, v) \qquad (3)$$

Here, $G(u,v)$ is the blurred image, $F(u,v)$ is an uncorrupted original image, $H(u,v)$ is the blurring function, which does the blurring and $N(u,v)$ is the additive noise term. All these terms are represented using frequency domain notations.

**2.1.2 Image noise model.**   The image degradation model, as discussed in an earlier section, consists of additive noise terms, which symbolize the possession of one of the several types of noise. This noise may be available at the time of image acquisition. This additive term appropriately models the noise in Charged Coupled Devices (CCD) cameras. When a CCD camera is used for imaging, multiple noise sources are present in the surroundings. One of these sources is modeled in this work and referred to as Gaussian noise. The Gaussian distribution models this noise as given in Eq 4 [20].

$$\eta(x, y) = \frac{1}{2\pi\sigma^2} e^{\left(-\frac{x^2+y^2}{2\sigma^2}\right)} \qquad (4)$$

Here $\sigma$ = Standard Deviation of Gaussian distribution.

In this work, the Peak Signal-to-Noise Ratio (PSNR) metric is used to define the noise variance added to blurred images.

## 2.2 Segmentation-based fractal texture features

Fractal dimension is a widely used texture measure. It indicates the self-similarity at several scales of an object pattern. For, if bounded set S is the union of t distinct copies of itself, then S will be self-similar, here a proportion of scales down each copy p. The fractal dimension ($f_d$) defined by Eq 5 is given below:

$$f_d = \frac{\log t}{\log p} \qquad (5)$$

We can estimate the roughness and irregularity in the object's surface through fractal dimension. The higher value of the fractal dimension indicates a coarser texture.

The Segmentation-based Fractal Texture Features algorithm is an efficient method for the extraction of texture. It consists of 2 main steps. In the first step, gray-level images are decomposed into 2*n of two-level images by the Two-Threshold Binary Decomposition (TTBD) algorithm, where n indicates the number of the threshold. In the 2nd step, decomposed binary images are utilized to calculate fractal dimension, area, and mean gray level. Due to this feature of SFTA, it is used in texture classification and object detection. Its algorithm is described below [21]:

```
Step1: multi-level Otsu thresholding algorithm
Step2: Th^C:={(th: th_i, th_{i+1}): th_i, th_{i+1}∈Th and i∈[1.…......|Th|-1]
Step3: Th^P:={(th: th_i, l_max): th_i∈Th and i∈[1.…......|Th|-1]}
Step 4: j: = 1.
for i: = 1:n
I_C:=TTS(I,t). where t∈Th^C
// Binary Decomposition for input image I
IP:=TTS(I, t). where t∈Th^P
∂^C(x,y):=FB(I_C)
∂^P(x,y):=FB(I_P)
FE[j]:=Box_Counting(∂^C)
FE[j + 1]:=Mean_Gray_Level(∂^C)
```

$$FE[j+2] := Pixel\_Count(\partial^C)$$
$$FE[j+3] := Box\_Counting(\partial^P)$$
$$FE[j+4] := Mean\_Gray\_Level(\partial^P)$$
$$FE[j+5] := Pixel\_Count(\partial^P)$$
```
j: = j + 6
end for
```

## 2.3 Segmentation-based fractal texture features

The statistical features are used to fetch the parameters like mean, skewness, standard distribution, entropy, etc., of textured images. Let $\lambda_i$ be a discrete random variable, which signifies diverse levels in a map, and let $p(\lambda_i)$ be the Probability Density Function (PDF). A histogram indicates the probability of occurrence of values $\lambda_j$ as measured by $p(\lambda_j)$. The feature set of every histogram has the following six features as defined in Eqs (6–11) [22].

- Mean- mean calculates the average value.

$$m = \sum\nolimits_{i=0}^{L-1} \lambda_i p(\lambda_i) \tag{6}$$

- Standard deviation- It calculates the mean contrast.

$$\sigma = \sqrt{\sum\nolimits_{i=0}^{L-1} (\lambda_i - m)^2 p(\lambda_i)} \tag{7}$$

- Smoothness—Smoothness calculates the relative smoothness of the gray intensities of a particular segment.

$$R = 1 - 1/(1 + \sigma^2) \tag{8}$$

- Skewness–Skewness computes the symmetry of the distribution.

$$S = \sum_{i=0}^{L-1} (\lambda_i - m)^3 p(\lambda_i) \tag{9}$$

- Uniformity–Uniformity is also known as energy.

$$U = \sum_{i=0}^{L-1} p^2(\lambda_i) \tag{10}$$

- Entropy–Entropy is a measure of randomness.

$$E = \sum_{i=0}^{L-1} p(\lambda_i) log\, p(\lambda_i) \tag{11}$$

## 2.4 Multilayer Feed Forward Neural Network and Back Propagation training algorithm

An Artificial Neural Network (ANN) or a Neural Network (NN) is a biologically inspired machine-learning algorithm. A typical neural network structure consists of an interconnected network of ordinary processing units. It provides a robust data modeling mechanism that is used to establish complex input and output relationships. The motivation for the growth of neural network computing developed from the aspiration to design an unconventional method of computing and to understand the processing of human intelligence. NN identically processes data to what the human brain does. The network architecture consists of a massive number of organized processing elements termed neurons. These neurons perform in a parallel manner to provide the solution to a particular problem. The neural network model works on the principle of learning by example. These examples must be chosen cautiously; otherwise, the model may work incorrectly. The problem is that there is no way to recognize whether the framework is decent, except if an error happens [23].

The structure of a neural network consists of neurons as building blocks. A neuron performs similarly to what the biological one does. It receives multiple data sources with distinct weights and has one output, which depends upon the information sources. A biological neuron can either 'fire' or not "fire" (when a neuron fires, it yields a beat of a couple hundred Hz). In an artificial neuron, 'firing' is typically spoken by a consistent one and not 'firing' by a zero. There is a wide range of sorts of neural systems now being used. The methods vary from each other in their design and the preparation calculations.

There are two principal sorts of learning: supervised and unsupervised learning. Supervised learning, for example, with Multilayer Perceptron (MLP), implies that the neural system knows the ideal yield and that the altering of weight coefficients is done in such a way that the determined and required outcomes are as close as could be expected under the circumstances. Unsupervised preparation of, for example, a Kohonen neural network implies that the ideal product isn't known; the framework is given a gathering of certainties (samples) and afterward left to itself to settle down (or not) to a steady state in some number of repetitions.

The abilities of Multilayer Feed Forward Neural Network (MLFNN) derive from the non-linearities available within the neuron unit. Every neuron in the network accepts inputs from former neurons in the network or accepts inputs provided externally (referred to as bias). The yields of the neurons are associated with different neurons or with the outside world. Each piece of information is associated with the neurons by weight. The neuron ascertains the weighted aggregate of the sources of information (referred to as activation), which goes through a non-straight exchange capacity to create the actual yield for the neuron. The most applied activation functions are of the sigmoidal kind. A joint Back Propagation Neural Network (BPNN) consists of an input layer, one hidden layer, and an output layer [24].

Among the calculations used to perform supervised training, the backpropagation calculation has developed as the most widely utilized and fruitful calculation for the design of feedforward systems. In this mode, the genuine yield of a neural system is contrasted with the ideal outcome. Loads, which are typically set haphazardly in any case, are then balanced by the system with the goal that the following emphasis, or cycle, will create a closer match between the ideal and the genuine yield. There are two unmistakable stages of the activity of back-propagated learning: the forward stage and the regressive stage. In the forward step, the information signals spread through the system layer by layer, in the end creating some reaction at the system's yield. The real response created is contrasted with the ideal reaction, producing blunder flags that are then proliferated in a regressive way through the system. In this retrogressive

period of activity, the parameters of the system are balanced to limit the total of the squared blunders. The steps in the BPN calculation are given as [25]:

*Step 1*: Randomly weight initialization.

*Step 2*: Till the termination condition remains false, repeat steps 3 to 10.

*Step 3*: For every training pair $x$: $t$, do steps 4 to 9.

*Step 4*: Every input unit $X_i$, $i = 1,2,3...,n$ receives the input signal, $x_i$, and broadcasts it to the next layer.

*Step 5*: For every neuron of the hidden layer ($Z_j$) where $j = 1,2,3...,p$.

$$z_{inj} = v_{oj} + \sum_i x_i v_{ij}$$

$$z_j = f(z_{inj})$$

broadcast $Z_j$ to the next layer. Where $v_{oj}$ is the bias on j$^{th}$ hidden unit.

*Step 6*: For every neuron of output layer $Y_k$, $k = 1,2,...,m$

$$Y_{ink} = w_{ok} + \sum_j z_j w_{jk}$$

$$y_k = f(y_{ink})$$

*Step 7*: Calculate $\delta_k$ for every output neuron,

$$\delta_k = (t_k - y_k)f'(y_{ink})$$
$$\Delta w_{jk} = \alpha \delta_k z_j$$
$$\Delta w_{ok} = \alpha \delta_k \qquad \qquad as\ (z_0 = 1)$$
$$Y_k$$

where $\delta_k$ is the portion of error correction weight adjustment for $w_{jk}$ i.e. due to an error at the output unit $y_k$, that is back propagated to the hidden unit, which passed to the previous unit $y_{k-1}$ and $\alpha$ is the learning rate.

*Step 8*: For every neuron of hidden layer

$$\delta_{inj} = \sum_{k=1}^m \delta_k w_{jk} \quad j = 1, 2, ....p$$
$$\delta_j = \delta_{inj} f'(z_{inj})$$
$$\Delta v_{ij} = \alpha \delta_j x_i$$
$$\Delta v_{oj} = \alpha \delta_j$$

Where $\delta_j$ is the part of error adjustment on weight modification for $v_{ij}$ i.e. owing to the backpropagation of error to the hidden unit $z_j$

*Step 9*: Revise the weights.

$$w_{jk}(new) = w_{jk}(old) + \Delta w_{jk}$$
$$v_{ij}(new) = v_{ij}(old) + \Delta v_{ij}$$

*Step 10*: Stop if reached to a specific error level.

In the case of supervised learning of the artificial neural network, it goes through learning before application. During learning, input and output data are provided to the network. This input and output data together form the learning data set. Training sets are required to be of enough size to contain all the required information. The test data set is applied to the network after finishing the learning. Testing is crucial to validate the network performance. It is used to understand the behavior of the trained network with unseen data. If a trained network does not give realistic outputs, it resembles that the network has not generalized.

## 2.5 Multilayer Feed Forward Neural Network and Back Propagation training algorithm

To carry out the texture classification, features are extracted from input images. Before feature extraction, images are resized. These features are passed to the neural network classification model for learning and labeling. The overall process is represented in Fig 1.

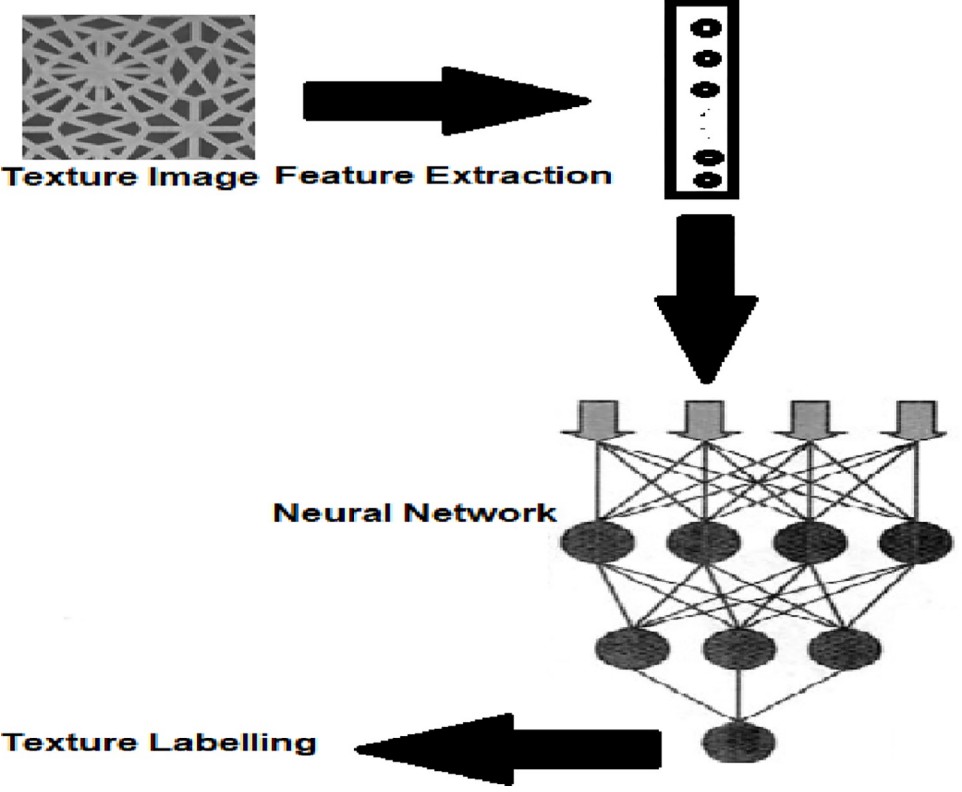

**Fig 1. A texture classification framework.**

## 2.6 Image dataset

The classifier models are evaluated on natural textures from Kylberg Texture Dataset v. 1.0 [26]. Fig 2 shows examples of texture images in the Kylberg Texture Dataset. This dataset comprises 28 classes of natural textures, which are macro photographs of real-world surfaces. Each class has 1920 patches of gray-scale images normalized with an average value of 127 and a standard deviation of 40. The patches have a resolution of 576×576 pixels and are resized into 256×256 pixels for feature extraction. To conduct the experiments in the presence of noise, 40 dB PSNR Gaussian Noise is synthetically introduced into all images. Work is implemented using image processing and neural network toolbox available in Matlab 6.5. as shown in Fig 2.

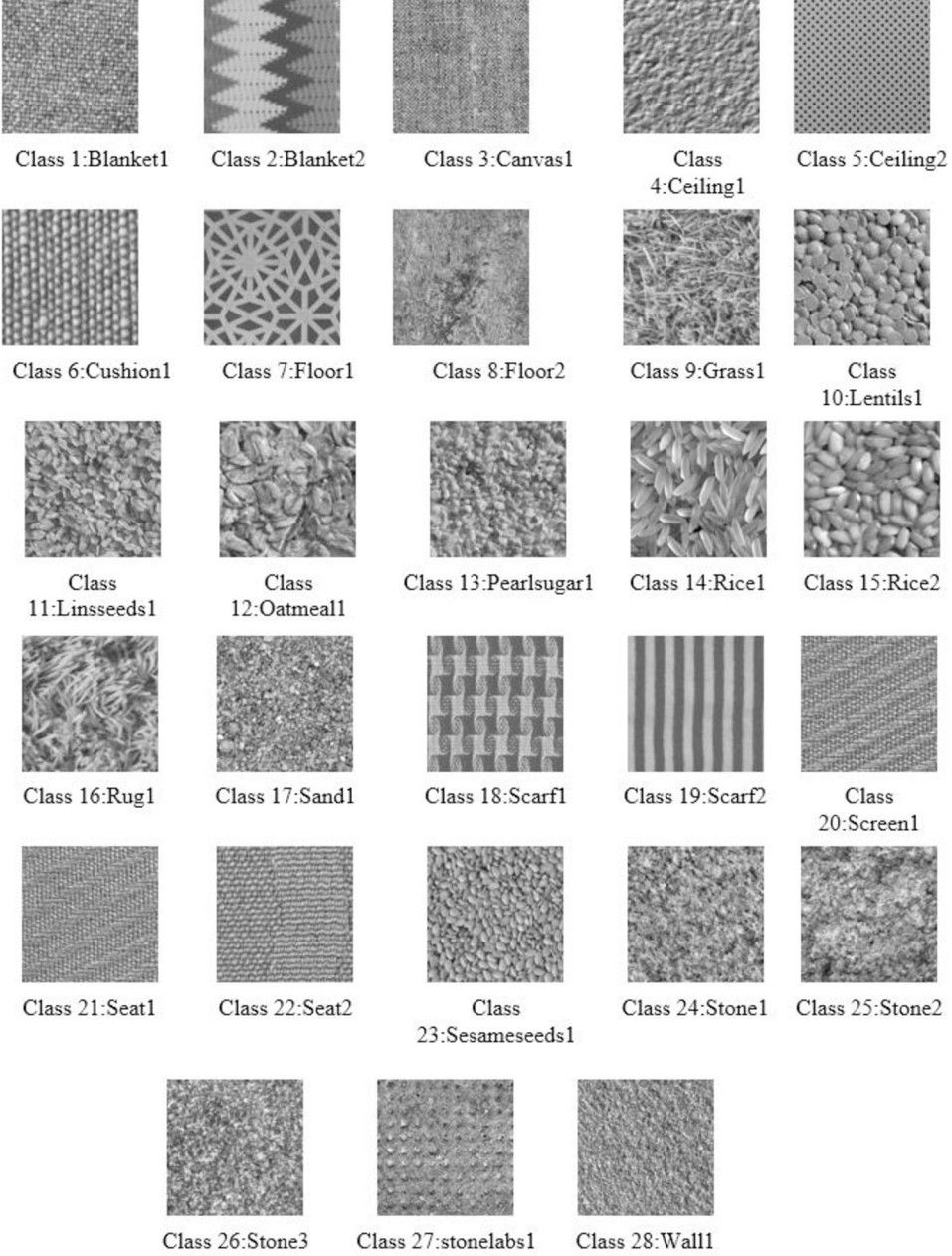

**Fig 2. Sample images from Kylberg texture image dataset.**

## 3. Experiment & results analysis

A feed-forward neural network with backpropagation as the network training algorithm is used to make the classification models. The extracted features are used to make the training dataset. The learning and testing feature set is standardized in the [0, 1] range, while the output class is allocated to logical zero for minimum probability and logical one for maximum likelihood. The sigmoid function is used as the transfer function for the hidden layers. The count of neurons in the input layer is the same as the count of features extracted from the dataset of the image, which is twenty-four for the first model with SFTF features and six for statistical texture features. Learning and testing of neural networks are carried out with a different number of hidden layers, which will help in finding the optimum count of hidden layers. One hidden layer comprising 10 neurons gave the best performance in the final architecture. Twenty-eight neurons are used in the output layer, corresponding to the number of classes in the image database. To prevent early stopping, data is divided into training sets with a 50% share, validation sets with a 20% share, and testing develops with a 30% share. After each epoch, a validation error is recorded. With the start of overfitting, the validation set error also starts increasing. When the validation error exceeds a specific epoch count, the training procedure is terminated and a trained model having the minimum validation error at that epoch's level is used.

Statistical metrics, namely accuracy, sensitivity, specificity, precision, and F-score, are used to evaluate the classification model. The explanation of each metric is as follows [27, 28]:

**Precision:** It indicates about what %age of +ive classification is actual.

$$Precision = \frac{True\ Positive}{[\text{True Positive} + \textit{False Positive}] = \textit{Total Predicted Positive}}$$

**Sensitivity:** It indicates that between the total prediction, what %age of the +ive prediction is actual. It is also called recall.

$$Sensitivity = \frac{True\ Positive}{[True\ Positive + False\ Negative] = Total\ Actual\ Positive}$$

**Specificity:** It indicates that between the total prediction, what %age of -ive predictions are actual. It is also called selectivity.

$$Specificity = \frac{True\ Negative}{[True\ Negative + False\ Positive] = Total\ Actual\ Negative}$$

**F-score:** It is the weighted mean of precision and recall to recall and precision.

$$F - Score = 2 * \frac{Recall * Precision}{Recall + Precision}$$

Table 1 presents the results for Segmentation-based fractal texture features with and without noise. The average accuracy of the model is 99%, and in the presence of Gaussian noise, the accuracy is reduced to 97%. Table 2 presents the results for the statistical moment-based texture analysis method in the absence and existence of Gaussian noise. The mean accuracy of the model is 98%, and in the presence of Gaussian noise, the accuracy is reduced to 95%.

## 4. Discussion

Sensitivity, accuracy, precision, specificity, and F-score are the most important parameters utilized to analyze the performance of any classifier model. Accuracy represents the correctness of the classifier in predicting the correct class [29]. The results are also compared with

**Table 1. Classification results with segmentation-based fractal texture analysis.**

| Texture Class | Model-based on SFTF | | | | | Model-based on SFTF with noisy images | | | | |
|---|---|---|---|---|---|---|---|---|---|---|
| | Accuracy | sensitivity | specificity | Precision | F-score | Accuracy | sensitivity | specificity | Precision | F-score |
| Blanket1 | 1.00 | 1.00 | 1.00 | 1.00 | 1.00 | 1.00 | 1.00 | 1.00 | 1.00 | 1.00 |
| Blanket2 | 0.99 | 0.98 | 0.99 | 0.94 | 0.96 | 0.97 | 0.64 | 0.98 | 0.61 | 0.62 |
| Canvas1 | 0.99 | 0.99 | 0.99 | 0.98 | 0.99 | 0.98 | 0.85 | 0.99 | 0.79 | 0.81 |
| Ceiling1 | 0.99 | 0.95 | 0.99 | 0.96 | 0.96 | 0.96 | 0.46 | 0.97 | 0.45 | 0.45 |
| Ceiling2 | 1.00 | 1.00 | 1.00 | 1.00 | 1.00 | 1.00 | 1.00 | 1.00 | 1.00 | 1.00 |
| Cushion1 | 1.00 | 1.00 | 1.00 | 1.00 | 1.00 | 0.97 | 0.72 | 0.98 | 0.67 | 0.69 |
| Floor1 | 0.96 | 0.35 | 0.98 | 0.50 | 0.41 | 0.96 | 0.15 | 0.99 | 0.55 | 0.23 |
| Floor2 | 0.96 | 0.65 | 0.97 | 0.50 | 0.56 | 0.96 | 0.88 | 0.96 | 0.47 | 0.62 |
| Grass1 | 0.99 | 0.93 | 0.99 | 0.94 | 0.94 | 0.95 | 0.11 | 0.98 | 0.25 | 0.16 |
| Lentils1 | 0.99 | 1.00 | 0.99 | 0.99 | 0.99 | 0.97 | 0.55 | 0.99 | 0.73 | 0.63 |
| Linsseeds1 | 0.99 | 0.98 | 0.99 | 0.97 | 0.97 | 0.95 | 0.43 | 0.97 | 0.42 | 0.42 |
| Oatmeal1 | 0.99 | 0.97 | 0.99 | 0.93 | 0.95 | 0.96 | 0.11 | 0.99 | 0.35 | 0.17 |
| Pearlsugar1 | 0.99 | 0.95 | 0.99 | 0.99 | 0.974 | 0.95 | 0.37 | 0.97 | 0.32 | 0.34 |
| Rice1 | 0.99 | 0.99 | 0.99 | 0.98 | 0.98 | 0.96 | 0.59 | 0.97 | 0.48 | 0.53 |
| Rice2 | 0.99 | 0.98 | 0.99 | 0.99 | 0.98 | 0.96 | 0.64 | 0.98 | 0.54 | 0.59 |
| Rug1 | 0.99 | 0.98 | 0.99 | 0.96 | 0.97 | 0.95 | 0.18 | 0.98 | 0.30 | 0.23 |
| Sand1 | 0.99 | 0.96 | 0.99 | 0.92 | 0.94 | 0.96 | 0.75 | 0.97 | 0.52 | 0.62 |
| Scarf1 | 1.00 | 1.00 | 1.00 | 1.00 | 1.00 | 0.96 | 0.70 | 0.97 | 0.51 | 0.59 |
| Scarf2 | 1.00 | 1.00 | 1.00 | 1.00 | 1.00 | 0.99 | 1.00 | 0.99 | 0.95 | 0.97 |
| Screen1 | 0.99 | 0.99 | 1.00 | 1.00 | 0.99 | 0.97 | 0.73 | 0.98 | 0.69 | 0.71 |
| Seat1 | 0.99 | 0.97 | 1.00 | 1.00 | 0.98 | 0.97 | 0.63 | 0.98 | 0.6 | 0.61 |
| Seat2 | 1.00 | 1.00 | 1.00 | 1.00 | 1.00 | 0.97 | 0.57 | 0.99 | 0.76 | 0.65 |
| Sesameseeds1 | 0.99 | 1.00 | 0.99 | 0.99 | 0.99 | 0.96 | 0.78 | 0.97 | 0.55 | 0.64 |
| Stone1 | 0.99 | 0.98 | 0.99 | 0.96 | 0.97 | 0.96 | 0.55 | 0.97 | 0.50 | 0.52 |
| Stone2 | 0.99 | 0.96 | 0.99 | 0.98 | 0.97 | 0.95 | 0.31 | 0.98 | 0.38 | 0.35 |
| Stone3 | 0.99 | 0.95 | 0.99 | 0.97 | 0.96 | 1.00 | 1.00 | 1.00 | 1.00 | 1.00 |
| Stonelabs1 | 0.99 | 0.98 | 0.99 | 0.99 | 0.98 | 0.97 | 0.68 | 0.98 | 0.67 | 0.67 |
| Wall1 | 0.99 | 0.9 | 0.99 | 0.93 | 0.91 | 0.97 | 0.58 | 0.98 | 0.66 | 0.62 |
| **Average** | **0.99** | **0.94** | **0.99** | **0.94** | **0.94** | **0.97** | **0.60** | **0.98** | **0.60** | **0.59** |

Convolutional Neural Networks. ConvNets are a subclass of neural networks that are mostly employed in speech and picture recognition applications. With no loss of information, its integrated convolutional layer lowers the high dimensionality of images. Convnets are, therefore, very well suited for texture classification. Five convolutional layers, four max-pooling layers, two fully connected layers, and a Softmax classifier output layer make up the ConvNet model being used. The activation function of the rectified linear unit is contained in the first convolution layer. It helps the model do better and acquire knowledge more quickly. Max pool layer and convolution are combined from layer one to layer five. A max pooling layer with a size of 2 x 2 and a stride of 2 is applied after each convolution layer. 3.0 kernel size and the number of filters applied. Fig 3 displays the comparison of different classifiers on the scale of accuracy. The segmentation-based fractal texture feature has given the best accuracy, as shown in Fig 3.

Precision is a measure of consistency. It represents the ability of the classifier to return only relevant cases. Fig 4 shows the comparison of different classifiers on the scale of precision. The segmentation-based fractal texture feature has the best value in terms of precision. Sensitivity represents the ability of the classifier to recognize all relevant cases correctly. Fig 5 shows the

**Table 2. Classification results with statistical moment-based texture analysis.**

| Texture Class | Model-based on SMBTF | | | | | Model-based on SMBTF with noisy images | | | | |
|---|---|---|---|---|---|---|---|---|---|---|
| | Accuracy | sensitivity | specificity | Precision | F-score | Accuracy | sensitivity | specificity | Precision | F-score |
| Blanket1 | 0.99 | 0.99 | 0.99 | 0.97 | 0.98 | 0.97 | 0.7 | 0.98 | 0.67 | 0.68 |
| Blanket2 | 0.99 | 0.98 | 0.99 | 0.98 | 0.98 | 0.96 | 0.56 | 0.97 | 0.49 | 0.52 |
| Canvas1 | 0.99 | 0.98 | 0.99 | 0.98 | 0.98 | 0.96 | 0.58 | 0.98 | 0.53 | 0.55 |
| Ceiling1 | 0.98 | 0.85 | 0.99 | 0.82 | 0.83 | 0.92 | 0.38 | 0.94 | 0.21 | 0.27 |
| Ceiling2 | 1.00 | 1.00 | 1.00 | 1.00 | 1.00 | 0.99 | 1.00 | 0.99 | 0.98 | 0.99 |
| Cushion1 | 0.99 | 0.93 | 0.99 | 0.88 | 0.90 | 0.95 | 0.45 | 0.97 | 0.42 | 0.43 |
| Floor1 | 0.96 | 0.57 | 0.97 | 0.5 | 0.53 | 0.96 | 0.39 | 0.98 | 0.50 | 0.44 |
| Floor2 | 0.96 | 0.41 | 0.98 | 0.48 | 0.44 | 0.95 | 0.49 | 0.97 | 0.38 | 0.43 |
| Grass1 | 0.98 | 0.78 | 0.99 | 0.82 | 0.80 | 0.95 | 0.03 | 0.99 | 0.14 | 0.05 |
| Lentils1 | 0.99 | 0.91 | 0.99 | 0.88 | 0.90 | 0.95 | 0.26 | 0.97 | 0.30 | 0.28 |
| Linsseeds1 | 0.98 | 0.83 | 0.98 | 0.70 | 0.76 | 0.93 | 0.11 | 0.96 | 0.11 | 0.11 |
| Oatmeal1 | 0.98 | 0.76 | 0.99 | 0.77 | 0.77 | 0.96 | 0.00 | 0.99 | 0.00 | 0.00 |
| Pearlsugar1 | 0.99 | 0.93 | 0.99 | 0.95 | 0.94 | 0.94 | 0.05 | 0.98 | 0.10 | 0.07 |
| Rice1 | 0.99 | 0.92 | 0.99 | 0.95 | 0.93 | 0.95 | 0.09 | 0.98 | 0.18 | 0.12 |
| Rice2 | 0.99 | 0.92 | 0.99 | 0.86 | 0.89 | 0.95 | 0.06 | 0.98 | 0.13 | 0.09 |
| Rug1 | 0.96 | 0.38 | 0.98 | 0.58 | 0.46 | 0.94 | 0.05 | 0.97 | 0.08 | 0.06 |
| Sand1 | 0.97 | 0.68 | 0.98 | 0.70 | 0.69 | 0.95 | 0.03 | 0.99 | 0.11 | 0.04 |
| Scarf1 | 0.99 | 0.97 | 0.99 | 0.99 | 0.98 | 0.92 | 0.43 | 0.94 | 0.23 | 0.30 |
| Scarf2 | 0.99 | 1.00 | 0.99 | 0.99 | 0.99 | 0.93 | 0.66 | 0.94 | 0.32 | 0.43 |
| Screen1 | 0.99 | 0.97 | 0.99 | 0.93 | 0.95 | 0.95 | 0.56 | 0.97 | 0.43 | 0.48 |
| Seat1 | 0.99 | 0.96 | 0.99 | 0.98 | 0.97 | 0.95 | 0.39 | 0.98 | 0.42 | 0.40 |
| Seat2 | 0.99 | 0.93 | 0.99 | 0.87 | 0.90 | 0.96 | 0.35 | 0.98 | 0.44 | 0.39 |
| Sesameseeds1 | 0.99 | 0.96 | 0.99 | 0.92 | 0.94 | 0.93 | 0.38 | 0.95 | 0.24 | 0.29 |
| Stone1 | 0.98 | 0.77 | 0.99 | 0.81 | 0.79 | 0.93 | 0.11 | 0.96 | 0.11 | 0.11 |
| Stone2 | 0.98 | 0.76 | 0.98 | 0.72 | 0.74 | 0.92 | 0.16 | 0.95 | 0.11 | 0.13 |
| Stone3 | 1.00 | 1.00 | 1.00 | 1.00 | 1.00 | 1.00 | 1.00 | 1.00 | 1.00 | 1.00 |
| Stonelabs1 | 0.99 | 0.98 | 0.99 | 0.97 | 0.97 | 0.95 | 0.5 | 0.97 | 0.44 | 0.47 |
| Wall1 | 0.98 | 0.76 | 0.99 | 0.82 | 0.79 | 0.94 | 0.26 | 0.97 | 0.25 | 0.25 |
| **Average** | 0.98 | 0.85 | 0.99 | 0.85 | 0.85 | 0.95 | 0.36 | 0.97 | 0.33 | 0.34 |

comparison of different classifiers on the scale of sensitivity. Segmentation-based fractal texture feature has the best value of sensitivity.

There is always a trade-off between recall and precision. Which parameter between precision and recall should be maximized depends on the problem. However, there is another metric through which we can consider both precision and recall at the same time, it is called the F-score. Instead of striking a balance between the two metrics, the aim is to maximize a single parameter, the F-score. The F-score is the harmonic mean of recall and precision. Fig 6 depicts the comparison of different classifiers on the scale of the F-score. Segmentation-based fractal texture feature has given the best value of the F-score, as shown in Figs 4 and 5.

Specificity tells us how many percentages of actual negative value have correctly identified as -ive value. Fig 7 shows the comparison of different classifiers on the scale of specificity. It has been found that the segmentation-based fractal texture feature has the best value of specificity.

The collective analysis concludes that segmentation-based fractal texture features have shown superior results to statistical moments-based texture features. It has been proven from the results that SFTF is robust not only in quality images but also in noisy images. In the

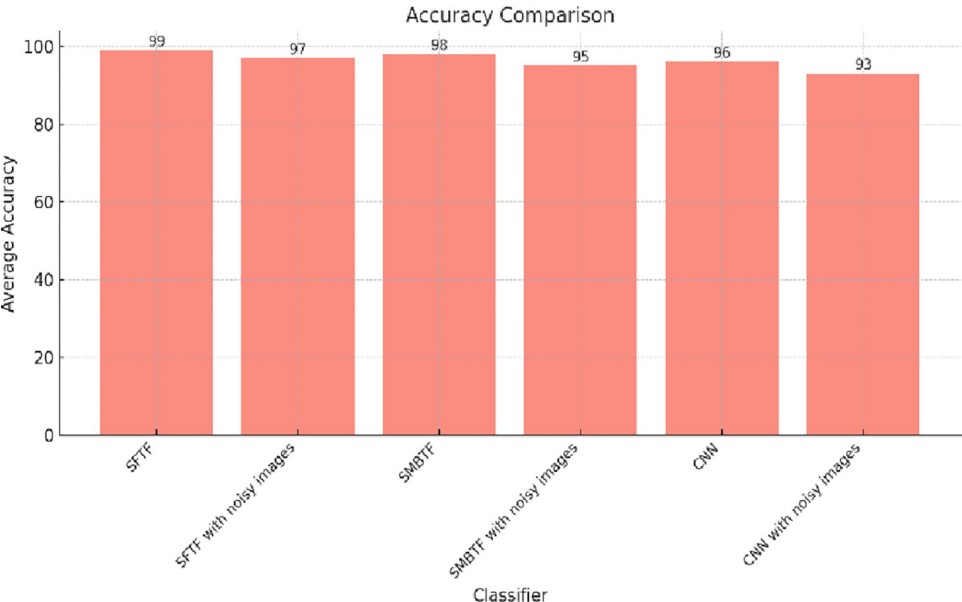

**Fig 3. Comparison of classifiers on the scale of accuracy.**

presence of Gaussian noise, the accuracy of the SMBTF-based model is reduced to 95% from 98%, whereas the accuracy of the SFTF model is just reduced to 97% from 99%. It confirms that SFTF features are more robust to noise than SMBTF features, as shown in Figs 6 and 7.

Table 3 presents a performance comparison of the proposed method with prominent work for classification on the Kylberg Texture Dataset. It is evident from the results that the proposed method consisting of Segmentation-Based Fractal Texture Features and Neural Network provides comparable performance in most of the ways.

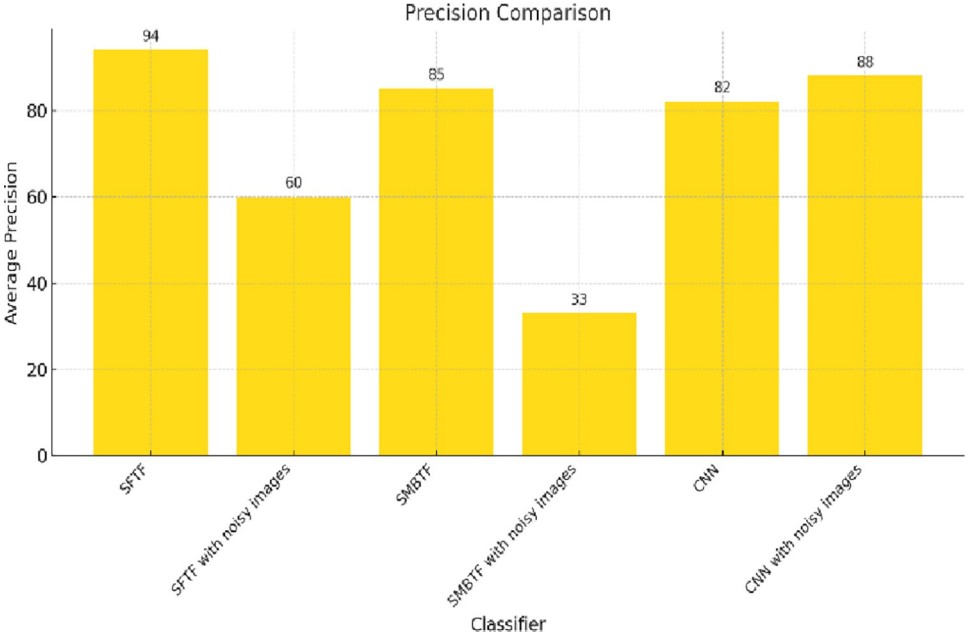

**Fig 4. Comparison of classifiers on the scale of precision.**

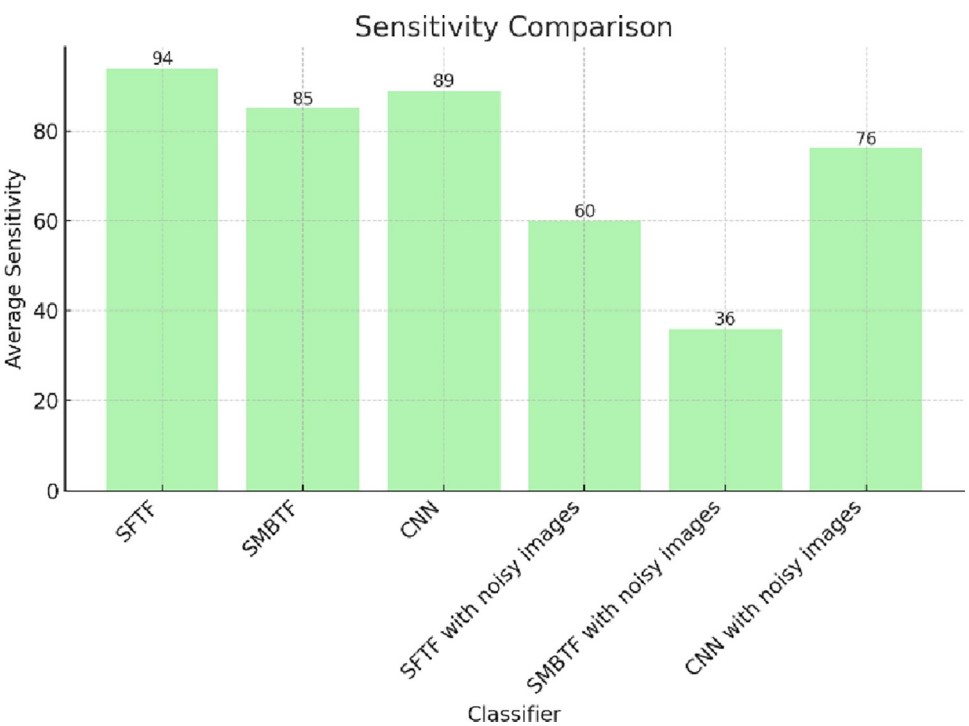

**Fig 5. Comparison of classifiers on the scale of sensitivity.**

## 5. Conclusions

Texture-based classification is one of the prominent methods for content-based image retrieval. Most different texture classifiers work in two stages. Its first phase is feature extraction, which generates a feature-measure-based classification of each texture class. Recognizing

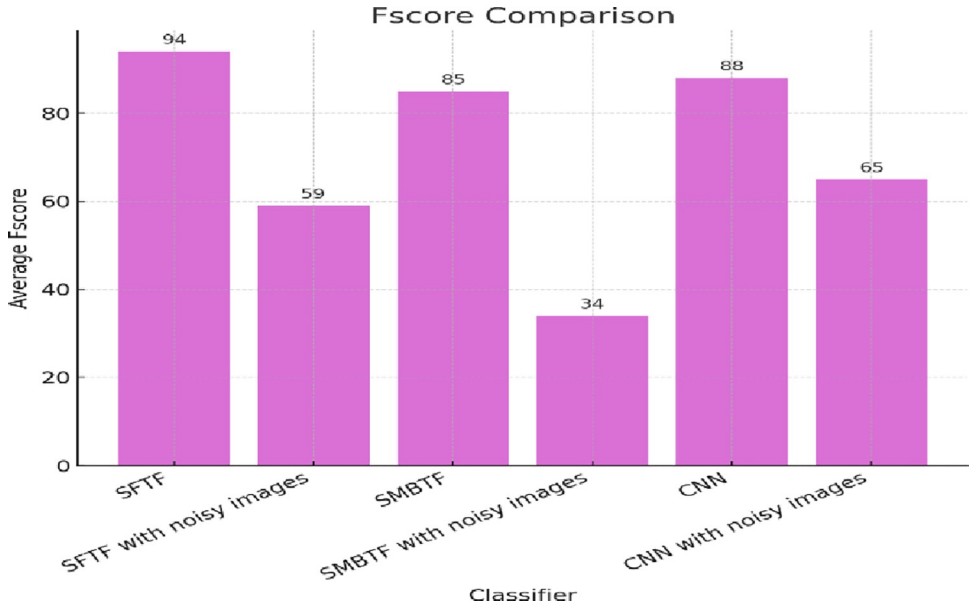

**Fig 6. Comparison of classifiers on the scale of F-score.**

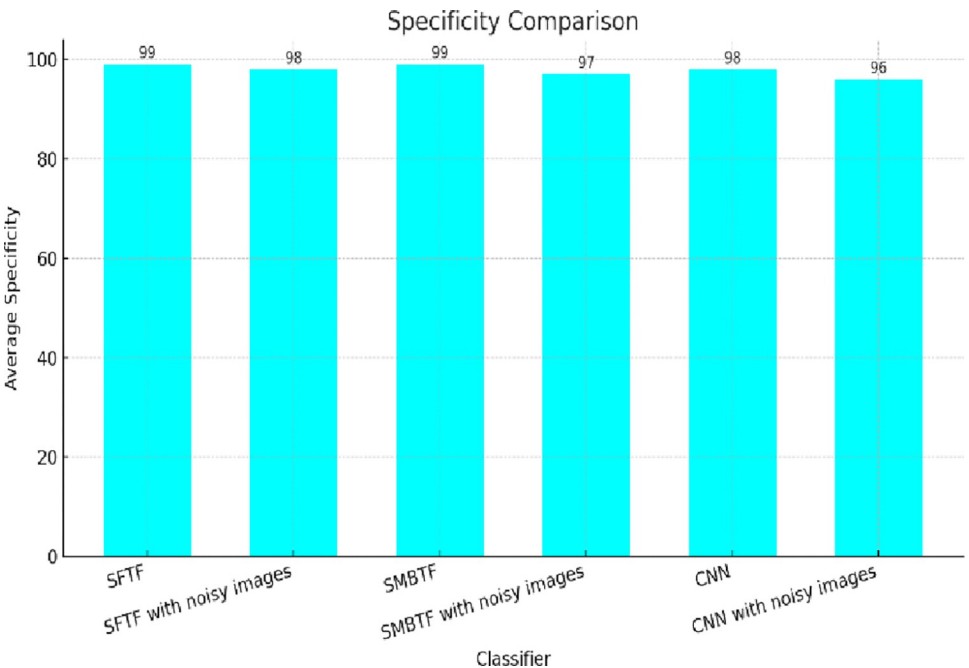

**Fig 7. Comparison of classifiers on the scale of specificity.**

and choosing distinctive features insensitive to irrelevant image transformations like translation, rotation, and scaling is essential. For similar textures, the metrics of selected features must presumably be comparable. However, designing a generalizable classifier is a challenging task, and most existing ones are application-specific and require varying degrees of domain expertise. However, these methods suffer from the existence of noise. It is of the utmost importance to propose and analyze a texture classification method in the presence of noise. The effect of noise is explored once the descriptor parameters have been tuned for each dataset in this work. Two different sets of texture features have been evaluated for a texture database of twenty-eight classes in the presence of noise. Two separate experiments are carried out for each stage of features. In the first experiment, no degradation is considered, while in the second experiment, Gaussian noise of 40 dB is regarded as a degradation factor. It is evident that the performance of the first model, which utilizes segmentation-based fractal texture features,

**Table 3. Performance comparison of the proposed method with prominent work for classification on Kylberg Texture Dataset [26].**

| Research work | Feature and Classifier | Accuracy |
|---|---|---|
| Agarwal et al. [30] | Convolutional Neural Network | 96.42% |
| Kaya et al. [31] | Directional local binary patterns and Support Vector Machine | 95.83% |
| Kaya et al. [31] | Directional Local Binary Patterns and Linear Discriminant Analysis | 93.33% |
| Kaya et al. [31] | Directional local binary patterns and k nearest neighbor | 95.47% |
| Kaya et al. [31] | Directional local binary patterns and Bayesian Network | 89.76% |
| Kaya et al. [31] | Directional local binary patterns and Random Forest | 94.64% |
| Kylberg et al. [32] | Local Binary Patterns and 1 Nearest-Neighbor | 97.8% |
| Kylberg et al. [32] | Gray-level co-occurrence matrix and 1 Nearest-Neighbor | 84.7% |
| *Proposed Method* | *Segmentation-Based Fractal Texture Features and Neural Network* | *99%* |

is better than the second model, which uses statistical moment-based texture analysis features. In the future, this texture categorization work can be extended by analyzing the performance in the presence of other image degradation factors such as blur view invariance, cluttered backgrounds, occlusion, and noise.

## Author Contributions

**Conceptualization:** Shamik Tiwari, Antima Jain, Hairulnizam Mahdin, Senthil Athithan.

**Data curation:** Shamik Tiwari.

**Formal analysis:** Shamik Tiwari, Deepak Gupta, Rahmat Hidayat.

**Funding acquisition:** Akhilesh Kumar Sharma, Izzatdin Abdul Aziz, Deepak Gupta, Antima Jain, Hairulnizam Mahdin.

**Investigation:** Shamik Tiwari, Deepak Gupta, Antima Jain, Hairulnizam Mahdin, Senthil Athithan.

**Methodology:** Shamik Tiwari, Akhilesh Kumar Sharma, Deepak Gupta.

**Project administration:** Akhilesh Kumar Sharma, Deepak Gupta, Antima Jain.

**Resources:** Akhilesh Kumar Sharma, Izzatdin Abdul Aziz, Antima Jain, Hairulnizam Mahdin, Rahmat Hidayat.

**Software:** Senthil Athithan.

**Supervision:** Izzatdin Abdul Aziz, Antima Jain, Hairulnizam Mahdin.

**Validation:** Izzatdin Abdul Aziz, Antima Jain, Hairulnizam Mahdin, Senthil Athithan.

**Visualization:** Izzatdin Abdul Aziz, Senthil Athithan.

**Writing – review & editing:** Akhilesh Kumar Sharma, Deepak Gupta, Hairulnizam Mahdin.

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
