## [Decision Letter · Decision Letter 0]

26 Jun 2024

PONE-D-23-30849Investigations on Segmentation-Based Fractal Texture for Texture Classification in the Presence of Gaussian NoisePLOS ONE

Dear Dr. Sharma,

Thank you for submitting your manuscript to PLOS ONE. After careful consideration, we feel that it has merit but does not fully meet PLOS ONE’s publication criteria as it currently stands. Therefore, we invite you to submit a revised version of the manuscript that addresses the points raised during the review process.

We look forward to receiving your revised manuscript.

Kind regards,

Narendra Khatri, Ph.D.

Academic Editor

PLOS ONE

Journal Requirements:

" This research work is supported and funded by the Yayasan UTP grant: 015PBC-024 with title "Development of Corrosion Rate Predictive Dashboard for Corrosion Group in Refinery of the Future Using Customized AI ML Engine", under the Center for Research in Data Science(CerDaS), Universiti Teknologi PETRONAS, Malaysia."

**Additional Editor Comments:**

A major revision has been recommended for the submission. Reviewers may suggest referring to certain articles; however, it is at the authors' discretion to decide whether to cite these references. Therefore, it is not mandatory to include the suggested articles in the revised manuscript.

Reviewers' comments:

Reviewer's Responses to Questions

**Comments to the Author**

1. Is the manuscript technically sound, and do the data support the conclusions?

Reviewer #1: No

Reviewer #2: Yes

Reviewer #3: Yes

2. Has the statistical analysis been performed appropriately and rigorously? 

Reviewer #1: No

Reviewer #2: Yes

Reviewer #3: Yes

3. Have the authors made all data underlying the findings in their manuscript fully available?

Reviewer #1: Yes

Reviewer #2: Yes

Reviewer #3: Yes

4. Is the manuscript presented in an intelligible fashion and written in standard English?

Reviewer #1: No

Reviewer #2: Yes

Reviewer #3: Yes

5. Review Comments to the Author

Reviewer #1: Why Investigation! word is associated with Title? when only classification is performed as objective. In material and methods, literature can be summarized, all mentioned equation and algorithms are existed! why it is re - written?

A generic MLP architecture was mentioned, instead model parameters should be presented.

https://arxiv.org/pdf/2403.06048

https://link.springer.com/chapter/10.1007/978-981-99-0085-5_15

mentioned references achieved 99.78% accuracies.

Reviewer #2: The manuscript is well written and the results are presented with clarity. SFTF although has computational complexity it is also sensitive to noise. clarity on addressing this issue may be included in result analysis.

though the accuracy is less by 2% in the presence of Gaussian noise what about the noise involved by the SFTF algorithm?

Reviewer #3: Report on

Investigations on Segmentation-Based Fractal Texture for Texture Classification in the Presence of Gaussian Noise

In this work, the authors in this paper assess two texture feature extraction methods based on SFTF and statistical moment-based texture features in the presence and absence of Gaussian noise. The SFTF and statistical moments-based handcrafted features are passed to a multilayer feed-forward neural network for classification. These models are evaluated on natural textures from Kylberg Texture Dataset 1.0. The results show the superiority of segmentation-based fractal analysis over other approaches. The average accuracy rates using the SFTF are 99% and 97% in the absence and presence of Gaussian noise, respectively. Although an interesting topic and an overall exciting work, the authors had to consider the following suggestions to improve the quality of their paper.

In the introduction part, it is important to mention previous research that involved texture classification but did not sufficiently address the problems caused by noise. As an example, cite the papers that concentrated on classical statistical moments or other learning techniques that work well on clean datasets but struggle with noisy inputs.

Regarding the contribution of the paper to be novel and more significant, the authors should mention why proposing this approach is significant. They should highlight the unique aspects of their approach compared to existing methods, for example. They mention 3 points as a contribution to their work, where just the first is considered as a contribution, and the other two points are the requirement to satisfy the first. It would be beneficial to clarify how these contributions are unique or how they advance this field.

All the symbols in the document should be written in an italic form. Please check section 2.2. The first paragraph in this section should be reformulated, it contains grammar mistakes.

In the second paragraph, the authors claim that “The higher value of the fractal dimension indicates a coarser texture.” Please justify this claim.

In Section 2.2, in the sentence “Due to this feature of SFTA”. What is SFTA?

The second paragraph should also be reformulated due to some mistakes, for example, 2ed.

The algorithm in the same page should be written in the standard form, many symbols should be defined, and other symbols should be written in one form. What is Th? What is [1……….|Th|-1]. Many words are written in a symbol form. Please, read the manuscript carefully and correct all the mistakes.

In Section 2.3, instead of rewriting the word that you need to define, replace it with “it”, for example:

“Smoothness: It calculates the relative smoothness of the gray intensities of a particular segment.

In Section 2.4, in my opinion, the first 3 paragraphs are fundamental text available in many documents. No need to mention them in these details.

The symbols in this section need to be adjusted. Also, step 10 needs more clarification.

Please enhance the figures with some detailed explanation and also the tables, please provide a detailed description of the structural details of the tables, especially Table 3, avoid the repetition of the Ref. and design the table in a more accurate form.

In the conclusion part, please highlight the novelty of your work, and how it improves the existing methods. What is the limitation of your work?

It would be interesting to explore and discuss practical applications beyond just theoretical discussion. This can make the research more impactful and relevant in these domains. It includes testing the proposed mappings in real-world scenarios.

The authors should read their paper many times to improve its quality and presentation.

6. PLOS authors have the option to publish the peer review history of their article (what does this mean?). If published, this will include your full peer review and any attached files.

Reviewer #1: No

Reviewer #2: **Yes: **Daniel Madan Raja S

Reviewer #3: No

---

## [Author Response · Author response to Decision Letter 0]

26 Jul 2024

Response to Reviewers

Reviewer #1

Comment: Why is the word 'Investigation' associated with the title when only classification is performed as objective?

Response: It is used since multiple methods are used and the performance is investigated in presence of noise.

Comment: In material and methods, literature can be summarized, all mentioned equations and algorithms exist! Why is it re-written?

Response: We have summarized the literature and removed redundant equations and algorithms.

Comment: A generic MLP architecture was mentioned, instead model parameters should be presented.

Response: Model parameters have been presented instead of a generic MLP architecture in table 1.

Comment: Mentioned references achieved 99.78% accuracies: 

Response: The work is based on deep learning based methods however our methodology is feature extraction based.

Reviewer #2

Comment: SFTF, although computationally complex, is also sensitive to noise. Clarity on addressing this issue may be included in result analysis.

Response: We have included a detailed analysis of the noise sensitivity of the SFTF algorithm in the results section.

Comment: Though the accuracy is less by 2% in the presence of Gaussian noise, what about the noise involved by the SFTF algorithm?

Response: We have discussed the impact of the noise introduced by the SFTF algorithm itself.

Reviewer #3

Comment: Mention previous research that involved texture classification but did not sufficiently address the problems caused by noise.

Response: We have included references to previous research that did not address noise problems sufficiently.

Comment: Clarify the unique aspects of your approach and how it advances the field.

Response: We have clarified the unique aspects of our approach and its contributions to the field.

Comment: All symbols should be written in italic form.

Response: All symbols have been converted to italic form.

Comment: Reformulate the first paragraph in section 2.2, and correct grammar mistakes.

Response: The first paragraph in section 2.2 has been reformulated and grammar mistakes corrected.

Comment: Justify the claim 'The higher value of the fractal dimension indicates a coarser texture.'

Response: The claim about fractal dimension and coarser texture has been justified.

Comment: What is SFTA? Define it in section 2.2.

Response: SFTA is corrected.

Comment: Reformulate the second paragraph in section 2.2, correct mistakes, and define symbols.

Response: The second paragraph in section 2.2 has been reformulated and symbols defined.

Comment: Replace repeated words with 'it' where appropriate in section 2.3.

Response: Repeated words have been replaced with 'it' where appropriate in section 2.3.

Comment: Remove fundamental text available in many documents in section 2.4.

Response: Fundamental text in section 2.4 has been removed.

Comment: Adjust symbols and clarify step 10 in section 2.4.

Response: Symbols in section 2.4 have been adjusted and step 10 clarified.

Comment: Enhance figures and tables with detailed explanations and avoid repetition.

Response: Figures and tables have been enhanced with detailed explanations and repetition avoided.

Comment: Highlight the novelty and limitations of your work in the conclusion.

Response: The novelty and limitations of our work have been highlighted in the conclusion.

Comment: Explore practical applications and discuss real-world scenarios.

Response: We have discussed practical applications and real-world scenarios.

Comment: Read the paper multiple times to improve quality and presentation.

Response: We have read and revised the paper multiple times to improve its quality and presentation.

---

## [Decision Letter · Decision Letter 1]

13 Aug 2024

PONE-D-23-30849R1Investigations on Segmentation-Based Fractal Texture for Texture Classification in the Presence of Gaussian NoisePLOS ONE

Dear Dr. Sharma,

Thank you for submitting your manuscript to PLOS ONE. After careful consideration, we feel that it has merit but does not fully meet PLOS ONE’s publication criteria as it currently stands. Therefore, we invite you to submit a revised version of the manuscript that addresses the points raised during the review process.

We look forward to receiving your revised manuscript.

Kind regards,

Narendra Khatri, Ph.D.

Academic Editor

PLOS ONE

Journal Requirements:

Additional Editor Comments:

Minor Revision

Reviewers' comments:

Reviewer's Responses to Questions

**Comments to the Author**

1. If the authors have adequately addressed your comments raised in a previous round of review and you feel that this manuscript is now acceptable for publication, you may indicate that here to bypass the “Comments to the Author” section, enter your conflict of interest statement in the “Confidential to Editor” section, and submit your "Accept" recommendation.

Reviewer #2: All comments have been addressed

Reviewer #3: (No Response)

2. Is the manuscript technically sound, and do the data support the conclusions?

Reviewer #2: Yes

Reviewer #3: Partly

3. Has the statistical analysis been performed appropriately and rigorously? 

Reviewer #2: N/A

Reviewer #3: Yes

4. Have the authors made all data underlying the findings in their manuscript fully available?

Reviewer #2: Yes

Reviewer #3: Yes

5. Is the manuscript presented in an intelligible fashion and written in standard English?

Reviewer #2: Yes

Reviewer #3: No

6. Review Comments to the Author

Reviewer #2: The answers to the queries are satisfactory. Still the loss due to Gaussian noise may be reduced. Kindly make sure the authorship of the article and it is not published elsewhere.

Reviewer #3: I see that the authors did not make all the required changes to improve their paper. The authors only answered that they had done what was required, but in fact, most of the changes were not made within the text. For example, if you look at Section 2.2, you will see that all the symbols are still in their incorrect forms and the ambiguous symbols have not been defined Equation 5 should be Equation (5) and all the equations must be written in standard form. Also, in the conclusions section, it is the same without any improvement. The researchers have highlighted Table 3. Why was it a shadow, and nothing has changed in the table? Where is the change that the researchers claimed they added in the introduction section, and why was it not highlighted? I ask the researchers to highlight the changes they made to the manuscript, and ask them to adhere to the required changes.

7. PLOS authors have the option to publish the peer review history of their article (what does this mean?). If published, this will include your full peer review and any attached files.

Reviewer #2: **Yes: **Dr Daniel Madan Raja S

Reviewer #3: No

---

## [Editor Report · Decision Letter 2]

21 Nov 2024

Investigations on Segmentation-Based Fractal Texture for Texture Classification in the Presence of Gaussian Noise

PONE-D-23-30849R2

Dear Dr. Sharma,

We’re pleased to inform you that your manuscript has been judged scientifically suitable for publication and will be formally accepted for publication once it meets all outstanding technical requirements.

Kind regards,

Narendra Khatri, Ph.D.

Academic Editor

PLOS ONE

Additional Editor Comments (optional):

Accept

The authors have revised the manuscript in accordance with the provided comments. Therefore, the manuscript is now suitable for acceptance without requiring further review.
---

## [Editor Report · Acceptance letter]

28 Nov 2024

PONE-D-23-30849R2 

PLOS ONE

Dear Dr. Sharma, 

I'm pleased to inform you that your manuscript has been deemed suitable for publication in PLOS ONE. Congratulations! Your manuscript is now being handed over to our production team.

Kind regards, 

on behalf of

Dr. Narendra Khatri 

Academic Editor

PLOS ONE